# Orthogeriatric co-management for older patients with a major osteoporotic fracture: Protocol of an observational pre-post study

Sigrid Janssens[1], Marian Dejaeger[1,2], An Sermon[3,4], Katleen Fagard[1,2], Marie Cerulus[2,5], Heidi Cosyns[2,6], Johan Flamaing[1,2], Mieke Deschodt[1,5]*

1 Department of Public Health and Primary Care, Gerontology and Geriatrics, KU Leuven, Leuven, Belgium, 2 Department of Geriatric Medicine, University Hospitals Leuven, Leuven, Belgium, 3 Department of Development and Regeneration, KU Leuven, Leuven, Belgium, 4 Department of Traumatology, University Hospitals Leuven, Leuven, Belgium, 5 Competence Centre of Nursing, University Hospitals Leuven, Leuven, Belgium, 6 Department of Health, University Colleges Leuven, Leuven, Belgium

* mieke.deschodt@kuleuven.be

## Abstract

**Data Availability Statement:** No datasets were generated or analysed during the current study. All

### Background

Osteoporotic fractures are associated with postoperative complications, increased mortality, reduced quality of life, and excessive costs. The care for older patients with a fracture is often complex due to multimorbidity, polypharmacy, and presence of geriatric syndromes requiring a holistic multidisciplinary approach based on a comprehensive geriatric assessment. Nurse-led geriatric co-management has proven to prevent functional decline and complications, and improve quality of life. The aim of this study is to prove that nurse-led orthogeriatric co-management in patients with a major osteoporotic fracture is more effective than inpatient geriatric consultation to prevent in-hospital complications and several secondary outcomes in at least a cost-neutral manner.

### Methods

An observational pre-post study will be performed on the traumatology ward of the University Hospitals Leuven in Belgium including 108 patients aged 75 years and older hospitalized with a major osteoporotic fracture in each cohort. A feasibility study was conducted after the usual care cohort and prior to the intervention cohort to measure fidelity to the intervention components. The intervention includes proactive geriatric care based on automated protocols for the prevention of common geriatric syndromes, a comprehensive geriatric evaluation followed by multidisciplinary interventions, and systematic follow-up. The primary outcome is the proportion of patients having one or more in-hospital complications. Secondary outcomes include functional status, instrumental activities of daily living status, mobility status, nutritional status, in-hospital cognitive decline, quality of life, return to pre-fracture living situation, unplanned hospital readmissions, incidence of new falls, and mortality. A process evaluation and cost-benefit analysis will also be conducted.

                                                                                    

relevant data from this study will be made available upon study completion.

**Funding:** KF received a competitive grant by the University Hospitals Leuven to fund the G-COMAN programme. The G-COMAN Fractures study was funded with internal funding of the research unit Gerontology and Geriatrics, Department of Public Health and Primary Care of the Catholic University Leuven. The funders had and will not have a role in study design, data collection and analysis, decision to publish, or preparation of the manuscript.

**Competing interests:** The authors have declared that no competing interests exist.

## Discussion

This study wants to prove the beneficial impact of orthogeriatric co-management in improving patient outcomes and costs in a heterogenous population in daily clinical practice with the ambition of long-term sustainability of the intervention.

## Trial registration

International Standard Randomised Controlled Trial Number (ISRCTN) Registry: ISRCTN20491828. Registered on October 11, 2021, https://www.isrctn.com/ISRCTN20491828.

## Introduction

Osteoporosis is the most common musculoskeletal disease characterized by a low bone mass and disturbances in the bone microarchitecture composition [1, 2]. In combination with an increased fall risk, osteoporosis predisposes the older population to osteoporotic fractures [1, 2]. Osteoporotic fractures are fractures resulting from a low-energy trauma and classically occur at the femur, humerus, vertebrae, pelvis, and wrist [3]. In 2019, an estimated 4.3 million new osteoporotic fractures were reported in the European Union of which 100,000 in Belgium [4, 5]. With age being a well-known risk factor of osteoporosis, a major increase in osteoporotic fractures is expected in the coming decades due to population ageing [6, 7].

Osteoporotic fractures most often occur in a frail subgroup of geriatric patients having coexisting multimorbidity, polypharmacy, and geriatric syndromes [8–11]. As a consequence, these frail patients are more prone to develop complications, such as infections, thromboembolic events, delirium, and functional decline, which may result in a decreased quality of life and a high mortality rate [12–16]. One year after hip fracture surgery, 40% of preoperatively frail older patients are still unable to walk independently, 27% enter a nursing home for the first time, and 20% are deceased [17, 18]. Besides their impact on mortality and morbidity, osteoporotic fractures also have a huge socioeconomic impact. In 2019, the economic burden caused by osteoporotic fractures was estimated 57 billion euros in the European Union, which is an increase of 64% over almost one decade [4, 19].

A holistic multidisciplinary approach is indispensable for improving clinical outcomes of frail patients with a fracture [20]. Comprehensive geriatric assessment (CGA) is considered the golden standard to map underlying problems in order to prevent and detect complications and geriatric syndromes in hospitalized frail older patients [21]. CGA includes a multidimensional evaluation by a multidisciplinary care team to develop an individualized care plan with systematic follow-up. CGA can be delivered via different models of care [22]. A frequently applied care model for delivering care outside of the acute geriatric hospital ward is the use of mobile inpatient geriatric consultation teams, which act upon request of the non-geriatric team. However, research could not demonstrate a consistent impact of geriatric consultation teams on patient outcomes, which is hypothesized to be due to their recommendation-based and rather reactive character [23]. As a result, a shift in research and clinical practice towards geriatric co-management models has been observed [22]. Geriatric co-management is a proactive care model, focusing on both prevention and treatment of complications and geriatric syndromes, and characterized by shared responsibility and decision-making between the geriatric and non-geriatric care teams from admission until discharge. Geriatric-surgical co-

management models for surgical patients (i.e., orthopedic surgery, gastro-intestinal surgery, and an acute surgery unit) have shown a beneficial impact on length of stay, mortality, and hospital readmission rates [24]. Moreover, evidence revealed that geriatric patients with all types of osteoporotic fractures could improve from orthogeriatric co-management [25, 26].

In Belgian hospitals, there is currently no financing for systematic collaboration between geriatricians and surgeons. In general, the care for patients aged 75 years and over admitted to non-geriatric wards is organized and funded according to the Belgian 'Care Program for Geriatric Patients' (Royal Decree from 2007, updated in 2014) that provides inpatient geriatric consultation teams [27]. In 2017, the geriatrics department of the University Hospitals Leuven in Belgium used the available funding to reorganize its consultation team as a geriatric co-management team on the cardiology ward [28]. The new care model was found to be effective in improving in-hospital care processes, preventing functional decline and complications, and improving quality of life at 6-months post-discharge [29]. Based on these findings and the request of surgical wards for increased geriatric support, the geriatrics department of the University Hospitals Leuven decided to adapt the existing program on the cardiology ward to the older surgical population. This resulted in a geriatric-surgical co-management (named 'G-COMAN') program, which will be implemented successively on the traumatology, abdominal surgery, and vascular surgery ward in the University Hospitals Leuven.

In the current paper, we aim to provide a detailed overview of the methodology of the 'G-COMAN Fractures' study, which aims to evaluate the effectiveness of the geriatric co-management model on the traumatology ward. We hypothesize that orthogeriatric co-management for older patients with an osteoporotic fracture will have a beneficial impact on patient outcomes and inpatient costs. This paper is reported based on the 'Standard Protocol Items: Recommendations for Interventional Trials statement' in conjunction with the 'Template for Intervention Description and Replication' [30, 31].

## Materials and methods

### Study design and setting

The G-COMAN Fractures study is a single-center observational pre-post effectiveness study. This study will take place at the 56-bed traumatology ward of the University Hospitals Leuven, the largest university hospital in Belgium with 1995 beds and a level 1-trauma center. Patients in the pre-cohort will receive usual care, while patients in the post-cohort will receive the geriatric co-management intervention. Both groups will be assessed as presented in Fig 1.

The inclusion of patients in the usual care group will be done both prospectively and retrospectively. Due to the predefined strict implementation schedule of the G-COMAN intervention on the traumatology ward, the prospective inclusion of patients in the usual care group is limited to 5 months preceding the start of the implementation (October 25, 2021 –March 13, 2022). As this would result in insufficient power based on the sample size calculation, additional patients will be retrospectively included on a consecutive basis counting back from the start of the prospective inclusion of the pre-cohort. After the recruitment of the usual care group and the implementation of the G-COMAN intervention, but prior to the recruitment of the intervention group, a feasibility study was conducted to evaluate whether the core components of the G-COMAN intervention were successfully implemented. Once the core components were implemented, inclusion of the intervention group will be started. The expected duration for recruiting the post-cohort is estimated to be 10 months (Fig 2).

| | STUDY PERIOD | | | | | |
|---|---|---|---|---|---|---|
| | ADMISSION | DISCHARGE | FOLLOW-UP | | | |
| | $t_0$ | $t_1$ | $t_2$ | $t_3$ | $t_4$ | $t_5$ |
| Eligibility screen | X | | | | | |
| Informed consent | X | | | | | |
| Age, gender, comorbidities | X* | | | | | |
| Living situation | X* | X* | X | X | | X |
| Activities of Daily living (ADL) | X* | X* | X | X | X | |
| Instrumental ADL | X* | X | X | X | X | |
| Mobility status | X* | X | X | X | X | |
| Nutritional status | X* | | X | X | X | |
| Mental status | X* | X | | | | |
| Quality of life | X | X | X | X | X | X |
| Medication use | X* | X* | X | X | X | X |
| Falls and fracture history | X* | X* | X | X | X | X |
| Fracture and surgery details | | X* | | | | |
| Length of stay | | X* | | | | |
| In-hospital complications | | X* | | | | |
| Unplanned readmissions | | | X* | X* | | |
| Mortality | | X* | X* | | | X* |
| INTERVENTION: | | | | | | |
| Geriatric co-management | ←————————→ | | | | | |

**Fig 1. Overview of enrolment, intervention, and assessments of the G-COMAN Fractures study.** $t_0$ = admission; $t_1$ = discharge; $t_2$ = 1 month post-surgery; $t_3$ = 3 months post-surgery; $t_4$ = 6 months post-surgery; $t_5$ = 12 months post-surgery. * = retrospectively available using the patient's electronic health record.

## Study participants

Dutch-speaking patients aged 75 years and over who are admitted through the emergency department with a new osteoporotic fracture are included. Patients are eligible if they present with one of the following major osteoporotic fractures: fracture of the proximal femur, proximal humerus, pelvis (including acetabulum), thoracic and/or lumbar vertebrae, or distal radius and/or ulna. Patients are excluded if they suffer multiple fractures (except for presence of multiple vertebral fractures), a periprosthetic fracture, a pathological fracture (i.e., fracture caused

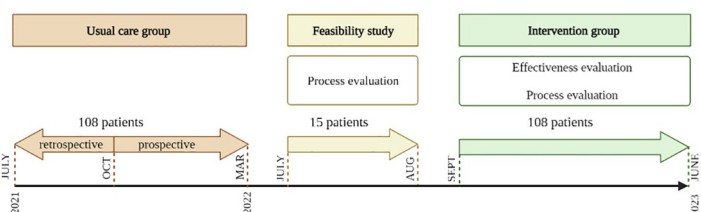

**Fig 2. Overview of the design and time frame of the G-COMAN Fractures study.**

by cancer), or a concomitant joint-infection (diagnosed by a physician), if they are transferred from another hospital or hospital ward (baseline data not available), or if they have an estimated life expectancy of less than 3 months at admission.

## Sample size

The sample size calculation was performed based on difference in risk of in-hospital complications. In the literature, a wide variation in in-hospital complications rates has been reported, ranging from 55% to 81.9% in the usual care group and from 19.9% to 57% in geriatric co-management [20, 32, 33]. We assumed a risk of in-hospital complications of 55% in the usual care group and 35% in the geriatric co-management group, hence an absolute reduction of 20%. With a significance level of 5%, a power of 80%, equal treatment groups and an expected loss of patients of about 10%, a total of 108 patients per group is needed. We will perform an interim analysis in the usual care group to evaluate the prevalence of the primary outcome (i.e., in-hospital complications) and if needed, adapt the number of included patients to guarantee sufficient power.

In the feasibility study, we aimed to recruit fifteen patients. This sample size was not based on a justified sample size calculation because there is little guidance as to the sample size required for a feasibility study [34]. However, we believed this number of patients would be sufficient to evaluate the feasibility of the intervention.

## Usual care

In the usual care group, the geriatric patient with a fracture is under the medical responsibility of a surgical resident who is supervised by a traumatologist and supported by a traumatology nurse specialist. The traumatology nurse specialist is an expert in both the nursing and medical field, such as diagnosing and treating patients, and takes care of the continuity of the care and treatment of the patients over the weekend. A multidisciplinary team is available for all patients on the traumatology ward. The multidisciplinary team consists of head nurses, nurse specialists, ward nurses, nurse aids, physiotherapists, occupational therapists, a psychologist, a speech therapist, a dietician, social workers, pharmacists, and a pastoral care service. All patient cases are being discussed in a weekly multidisciplinary team meeting. Surgical protocols for diagnosis and treatment are available for all patients admitted to the traumatology ward. Geriatric protocols for diagnosis and treatment have been issued for frail patients admitted with a proximal humerus or proximal femur fracture.

The geriatric consultation team is available upon active request of the surgical resident or traumatologist. This geriatric consultation team consists of geriatric nurses and occupational therapists and is supervised by a geriatrician. The geriatric consultation team conducts a multi-dimensional evaluation. In case of acute medical problems, a geriatric resident, supervised by a geriatrician, performs a bedside medical evaluation of the patient. During the daily briefings, the nurse of the geriatric consultation team and the geriatrician formulate tailored recommendations based on the identified problems. Recommendations are communicated in written form to the traumatology team and, if necessary, also orally. The traumatology team is responsible for implementing the recommended interventions. There is no systematic follow-up by the geriatric consultation team.

## G-COMAN intervention: Nurse-led orthogeriatric co-management

The G-COMAN intervention includes two main intervention components: (1) proactive geriatric care with automated protocols for all patients aged 75 years and over and (2) a risk

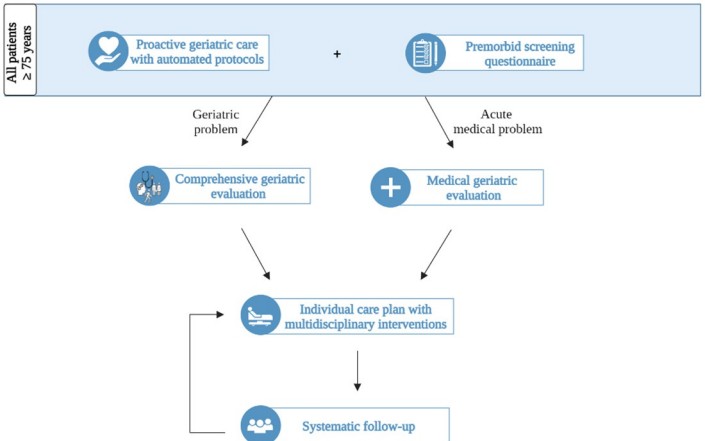

**Fig 3. Overview of the G-COMAN intervention.**

screening for all patients aged 75 years and over, followed by a nurse-led geriatric evaluation and multidisciplinary interventions with systematic follow-up in at risk patients (Fig 3).

First, all patients aged 75 years and older receive proactive geriatric care interventions that relate to their functional, somatic, psychological, or social status. To support this, various care plans focusing on those four domains were automatically programmed in the electronic health record. For example, each shift nurses see a care plan for monitoring the bowel transit of the patient.

Second, the premorbid situation, focusing on the functional, somatic, psychological, and social level, of all patients aged 75 years and older is documented via a screening questionnaire. This questionnaire is sent to the patient or their caregiver via the hospital's mobile application 'mynexuz.be' at admission to the traumatology ward. This is a secured application allowing patients or their caregivers to access their personal health record, including their appointments, medical reports, technical exams, as well as questionnaires sent out by healthcare professionals. As an alternative, the questionnaire may be offered via e-mail, a QR-code, or the interactive screens available in all hospital rooms. Validated scales are drawn from the answers to the questionnaire (e.g., Activities of Daily Living (ADL) Barthel scale [35], instrumental ADL (iADL) Lawton and Brody scale [36]). Furthermore, the answers may be a trigger for a bedside comprehensive geriatric evaluation to identify geriatric problems, geriatric syndromes, or geriatric risk factors. For example, if a risk for malnutrition is identified by the screening questionnaire, a comprehensive evaluation focusing on the nutritional status will be performed by a dietician. The results of the screening questionnaire and the bedside evaluation enable the development of a tailored interdisciplinary care plan during the weekly meeting with the multidisciplinary care team of the traumatology ward and by daily consultation between the G-COMAN traumatology nurse and the responsible ward nurse. A G-COMAN traumatology nurse, who is already employed on the traumatology ward, and a G-COMAN geriatric nurse were engaged for the G-COMAN intervention. Their role is to coach the multidisciplinary care team in their provision of proactive geriatric care (i.e., to motivate the healthcare professionals to execute the automated protocols and to coach them to provide the non-automated geriatric protocols), to facilitate the collaboration between the multidisciplinary care team, and to educate all healthcare professionals of the traumatology ward to make them aware of the complexity of geriatric patients and to facilitate their critical thinking about geriatric care. The care plan, based on the individual needs of the patients, is carried out by the head nurses,

nurse specialists, ward nurses, nurse aids, physiotherapists, occupational therapists, a psychologist, a speech therapist, a dietician, social workers, pharmacists, and a pastoral care service, supported by the G-COMAN traumatology nurse, G-COMAN geriatric nurse, and the geriatric resident. The surgical resident in collaboration with the geriatric resident is responsible for treating acute medical problems or geriatric syndromes (e.g., delirium, heart failure, pneumonia, . . .). Two times per week, the geriatric resident visits the geriatric patients on the traumatology ward in need of medical support.

## Variables

**Baseline variables.** Demographic variables include age, gender, and pre-fracture living situation (i.e., living at home alone or with partner or family member, living in an assisted living facility, or living in a nursing home). Clinical variables include body mass index (BMI), comorbidities based on the age-adjusted Charlson Comorbidity Index (CCI) [37], use of calcium-vitamin D supplements and/or anti-osteoporotic medication, presence of polypharmacy (i.e., ≥ 5 different medications), type of fracture, type of surgery, time to surgery, and American Society of Anesthesiology (ASA) score. Functional status will be measured with the 6-item Katz Index for ADL with a 3-point response scale (1 = independence; 2 = partial dependence; 3 = complete dependence) and the 10-item Barthel Index [35, 38]. iADL will be measured with the Lawton and Brody scale [36] and mobility status with the Parker mobility score [39]. Falls and fracture history will be questioned during a face-to-face patient interview and by consulting the electronic health record. Nutritional status will be evaluated using the Mini Nutritional Assessment (MNA)-short form [40]. Finally, the Mini-cog test, consisting of a three-item word memory and clock-drawing test, will be used to assess the cognitive status of the patient [41]. All baseline variables and their measurement methods are presented in Table 1.

**Outcome variables.** The primary outcome of the effectiveness study is the proportion of patients having one or more of the following in-hospital complications:

- Delirium: i.e., a state of acute and fluctuating confusion with disturbances on cognitive and attention level. The 4AT will be used three times a week to detect delirium. A score of 4 or more (ranging from 0 to 12) suggests delirium [42].

- Congestive heart failure: i.e., a functional or structural heart disorder resulting in reduced ventricular filling or ejection of blood to the systemic circulation. Congestive heart failure will be diagnosed in the presence of two major criteria or one major combined with two minor criteria based on the Modified Framingham Criteria [43].

- Pneumonia: defined by the World Health Organization (WHO) as an acute respiratory infection that affects the lungs caused by infectious agents, such as viruses, bacteria, or fungi. Diagnosis will be made by the treating physician.

- Deep venous thrombosis: i.e., the formation of one or more blood clots in one of the body's large veins. Diagnosis will be made through ultrasound imaging.

- Pulmonary embolism: i.e., a blood clot in the pulmonary arteries. Diagnosis will be made through computed tomography or nuclear imaging technics.

- Myocardial infarction: defined by the WHO as the demonstration of myocardial cell necrosis due to significant and sustained ischemia. The diagnosis is based on compatible findings on the electrocardiogram and blood enzymes.

**Table 1. Overview of baseline and outcome variables and their measurement methods.**

| Variable | Instrument | Description | Score | Source |
|---|---|---|---|---|
| Demographics | N/A | Age, gender | | Record |
| Comorbidities | Age-adjusted Charlson Comorbidity Index | 16 medical conditions scored based on severity and age | 3–37 No comorbidities = score 3 or 4 | Record |
| Living situation | N/A | Living at home (Alone or together) Assisted living Nursing home | | CRF |
| Activities of Daily living | Katz Index | Bathing, dressing, toileting, continence, feeding | 6–18 Independence = score 6 | CRF |
| | Modified Barthel Index | Bowels, bladder, grooming, toilet use, feeding, transfers, mobility, dressing, stairs, bathing | 0–100 Independence = score 100 | CRF |
| Instrumental ADL | Lawton and Brody Scale | Telephone use, shopping, food preparation, housekeeping, laundry, mode of transportation, medication use, finances | 0–8 Independence = 8 | CRF |
| Mobility status | Parker Mobility score | 3 questions related to mobility indoors, outdoors and to visit family | 0–9 No difficulties = score 9 | CRF |
| Nutritional status | Mini nutritional assessment | 6 screening questions | 0–14 Malnutrition = score 0–7 Risk of malnutrition = 8–11 | CRF |
| Mental status | Mini-cog | 3-item word memory and clock-drawing | 0–5 Impairment = score < 4 | Test |
| Quality of life | EQ-5D | Mobility, self-care, daily activities, pain, anxiety | | CRF |
| Medication use | N/A | Use of calcium/vitamin D and/or anti-osteoporotic medication Polypharmacy (≥ 5 medications) | | CRF Record |
| Falls and fracture history | N/A | Fall = "an unexpected event in which the patient comes to rest on the ground, floor or lower level" | | CRF Record |
| Fracture and surgery details | N/A | Type of fracture Type of surgery Time to surgery ASA score | | Record |
| Length of stay | N/A | Date of admission subtracted by date of hospital discharge | | Record |
| In-hospital complications | N/A | Delirium, congestive heart failure, pneumonia, deep venous thrombosis, pulmonary embolism, myocardial infarction, urinary tract infection, mortality | | 4AT Record |
| Unplanned readmissions | N/A | | | Record |
| Mortality | N/A | | | Record |

CRF = case report form

- Urinary tract infection: i.e., a bacterial infection of the urinary tract. The diagnosis is based on laboratory findings including one predominant bacterium > 100.000 and pyuria and treatment with antibiotics for at least three days [44].

- In-hospital mortality.

The secondary outcome variables include functional status, iADL status, mobility status, nutritional status, in-hospital cognitive decline, quality of life, return to pre-fracture living situation, unplanned hospital readmissions, incidence of new falls, and mortality.

Functional status will be assessed using the ADL scale [35, 38] on hospital admission and discharge, and at 1, 3 and 6 months post-surgery. iADL status will be evaluated using the Lawton and Brody scale [36] at hospital admission and discharge, and at 1, 3 and 6 months post-surgery. Mobility status will be determined based on the Parker mobility score [39] at hospital admission and discharge, and at 1, 3 and 6 months post-surgery. Nutritional status will be assessed with the MNA-short form [40] at hospital admission and at 1, 3 and 6 months post-surgery. In-hospital cognitive decline will be assessed by a decline in the Mini-cog score between hospital admission and discharge [41]. Quality of life will be assessed using the EQ-5D at hospital admission and discharge, and at 1, 3, 6 and 12 months post-surgery [45]. Return to pre-fracture living situation will be evaluated comparing the pre-fracture living situation

with the living situation at discharge and 1, 3, 6 and 12 months post-surgery. Length of hospital stay will be measured by subtracting the date of admission from the date of hospital discharge. Unplanned hospital readmissions, defined as being readmitted through the emergency department for any diagnosis, will be assessed at 30 and 90 days after discharge. Incidence of new falls will be evaluated at discharge and at 1, 3, 6 and 12 months post-surgery. Mortality will be evaluated at 30 days and 1-year post-discharge using the electronic health record that is linked to the federal database ('Kruispuntbank'). All outcome variables and their measurement methods are presented in Table 1.

Furthermore, a cost-benefit analysis will be performed using the hospital cost model recommended by the Belgian Health Care Knowledge Centre's report of 2010 [46, 47]. The in-hospital costs and benefits will be compared between the usual care group and the intervention group. Costs included are surgical procedures, use of medication, imaging, hospital stay, and costs for the co-management team. Revenues consist of conventional and private professional fees, rehabilitation convention fees, revenues from the geriatric day hospital and geriatric consultation team fees. Cost and income data will be delivered to our research team by the hospital administration.

## Process indicator variables

The process of care will be evaluated using process indicators both in the feasibility study and in the effectiveness study. In the feasibility study, the evaluation of the process indicators was used to assess how well the core components, and several other care processes, of the G-COMAN intervention were implemented as intended in the protocol prior to start of recruitment of the intervention group. The core components of the intervention needed to reach at least 80% of the patients to be feasible. In the effectiveness study (usual care vs. intervention group), the evaluation of the process indicators will determine how the process of care was changed as a result of the implementation of the G-COMAN intervention. The process indicators with their specified timing are indicated in Table 2.

## Data collection procedure

In the prospective usual care and intervention group, patients are recruited on hospital admission by the research assistant, who screens the patient's electronic health record for eligibility criteria and obtains written patient (or proxy) informed consent in a face-to-face interview. At the moment of recruitment, the research assistant will immediately complete the baseline case report form (CRF) in addition to consulting the electronic health record. During hospitalization, the research assistant will determine the incidence of complications by monitoring the patient record throughout hospitalization. Specifically for detecting delirium, a bedside 4AT assessment will be performed by the research assistant three times a week or until positive. Secondary outcomes at discharge will be assessed within 24 hours before hospital discharge during a face-to-face patient interview. At the moment of recruitment, the follow-up procedure after discharge will be discussed: a letter by post with the follow-up CRF or a telephone call. The follow-up assessments will take place at 1, 3, 6, and 12 months post-surgery (or post-admission in case of conservative treatment).

In the retrospective usual care group, the sample will be completed to the required sample size by recruiting consecutive patients who were admitted to the traumatology ward before 25 October 2021. The patient's electronic health record will be screened for eligibility. Baseline variables, in-hospital complications, and outcome variables at discharge will be collected using the electronic health record. As delirium could not be evaluated using the 4AT, the presence of delirium was established when a diagnosis was made by the geriatric consultation team stated

**Table 2. Overview of process indicators.**

| Process indicator | Timing |
|---|---|
| Core components: The proportion of patients who completed the screening questionnaire, who received a geriatric evaluation, and had an individual care plan with systematic follow-up | n/a |
| The proportion of patients who received physiotherapy | Within 24 hours of admission or postoperative |
| The proportion of patients who were evaluated using the food quadrant method | During 15 out of 21 meals per week during the hospitalization |
| The proportion of patients who received a swallowing screening | Within 24 hours of admission or postoperative |
| The proportion of patients without a urinary tract catheter if there is no indication for a urinary tract catheter | Within 24h postoperative |
| The proportion of patients without an intravenous catheter if there is no indication for an intravenous catheter | Within 48h postoperative |
| The proportion of patients that is free of a physical restraint if there is no indication for a physical restraint | n/a |
| The proportion of patients with a post-void residual volume of $\geq$ 300 ml that was removed the first time using intermittent catheterization and the second time using a urinary catheter | Before end of shift after detection of symptoms |
| The proportion of patients where the post-void residual bladder volume is monitored using a bladder scan after removal of urinary catheter | During the next shift after removal of catheter |
| The proportion of patients who received oral laxatives if they have not passed stool for 3 days | Before day 4 without stool |
| The proportion of patients who received a glycerin enema if they have not passed stool for 5 days | Before day 6 without stool |
| The proportion of patients where the Delirium Observation Screening (DOS) scale was completed | Every shift for 3 days from day 1 postoperative or if the patient is delirious |
| The proportion of patients with orientation measures in the patient's room (clock, calendar, or personal objects) | During the hospitalization |
| The proportion of patients with a pain evaluation | During every shift 3 days postoperative |
| The proportion of patients who received pain medication if the patient reported a pain score of 4 or higher (out of 10) | Within 1 hour of onset of symptoms |
| The proportion of patients who were re-evaluated if the patient reported a pain score of 4 or higher (out of 10) | Within 1 hour of onset of symptoms |
| The proportion of patients who were prescribed calcium/vitamin D supplements or anti-osteoporotic medication | At discharge |
| The proportion of patients who were referred to a fracture liaison service | At discharge |
| The proportion of patients who were referred to the geriatric day clinic | At discharge |

in the electronic patient record. Cognitive decline could not be determined in the retrospective cohort.

Due to the nature of the intervention and study design, health care professionals and patients cannot be blinded. Blinding of outcome assessors is not considered feasible due to limited resources for this study.

In the feasibility study, a research assistant recruited patients upon hospital admission and, after written patient (or proxy) informed consent was obtained, performed a face-to-face interview to be able to complete the baseline case report form (CRF) in addition to consulting the electronic health record. The process indicators (Table 2) were assessed during hospitalization by checking the electronic health records and by visiting the patients daily.

## Data analysis

Categorical data will be expressed as numbers and percentages. Continuous data will be expressed as means with standard deviations (normal distribution) or as medians and interquartile ranges (nonnormal distribution). The following baseline characteristics will be compared between the usual care group and intervention group to evaluate baseline differences: age, gender, BMI, comorbidities (Charlson Comorbidity Index), pre-fracture living situation, pre-fracture functional status (ADL), pre-fracture iADL status (Lawton and Brody), mobility status (Parker Mobility score), cognitive status (Mini-Cog), nutritional status (MNA-short form), type of fracture, and quality of life. They will be used to create propensity scores to control for potential baseline confounding. The propensity scores will be used to perform inverse probability of treatment weighting (IPTW) when comparing the primary and secondary outcomes between both groups [48]. The primary outcome will be compared with a two-sided (weighted) $\chi^2$ test with alpha set at 0.05. Secondary analyses will consist of (IPTW weighted) univariable statistical tests ($\chi^2$ test for categorical variables; t-test or Wilcoxon rank-sum test for continuous variables; log-rank and Gray's test for survival outcomes, the latter is required when mortality is a competing risk) to evaluate differences between the two treatment groups. For the secondary outcomes, a Bonferroni-Holm correction will be applied. In addition, univariable and multivariable logistic regression models will be used whereby the outcome will be the presence of any in-hospital complication. Nonparametric tests will be used to compare baseline characteristics between patients who are lost to follow-up from the control and intervention group. In the case of missing data, a multiple imputation analysis will be performed for baseline characteristics to be able to obtain the propensity scores [49]. Data analysis in the framework of this study will be performed in collaboration with the Leuven Biostatistics and Statistical Bioinformatics Centre.

## Ethics approval and consent to participate

The study was approved by the Ethics Committee Research UZ Leuven/KU Leuven (S65569). Every participant must give a written voluntary (proxy-)informed consent before the start of the study.

## Data management and monitoring

Study data will be collected and managed using REDCap®, a secured web application for building and managing electronic surveys and databases. The data will be pseudonymized. Every patient will receive a unique study number and there will be no combination of elements on the electronic CRFs that allows identification of the individual. Only the principal investigator (MaD) and the research assistant (SJ) will be able to link the data collected in REDCap® to the electronic health record using a subject identification log. The document will be stored separately and in a safe location by the principal investigator for 10 years, afterwards it will be deleted. The research assistant will introduce the data in REDCap®. The principal investigator will check for correct data collection.

## Dissemination plan

Scientific peer-reviewed publications are the fundamental route of dissemination of our findings. Articles will be published when possible as Open Access literature. We envision also several possible routes for disseminating the results to a broader non-expert audience including patient-oriented publications (e.g., regular lay press, UZ Magazine, . . .), lectures for lay-public, and forums for a wider medical and scientific audience (Artsenkrant, Belgisch tijdschrift

Ortho-Rheumato). At the end of the project a one-day symposium will be held, this symposium will also be open for the target population. Moreover, connections with local initiatives such as World Osteoporosis day can be used to disseminate the findings to the general public.

## Trial status

A total of 58 patients were included in the prospective usual care group from October 25, 2021, until March 13, 2022. In the retrospective usual care group, a total of 50 patients were included from July 25, 2021, until October 24, 2021. The feasibility study took place in the time frame between July–August 2022. The inclusion of patients in the intervention group started on September 5, 2022, and is expected to end on June 30, 2023.

## Discussion

This paper provides a detailed overview of the study design and methodology of the G-COMAN Fractures study. In this study, the feasibility and clinical effectiveness of a newly embedded orthogeriatric co-management intervention will be evaluated. We hypothesize that effectively implemented orthogeriatric co-management for older patients with any type of major osteoporotic fracture will decrease the incidence of in-hospital complications and will improve relevant clinical outcomes as ability to perform activities of daily living, mobility, and length of stay, and be at least cost-neutral.

Due to population ageing, the prevalence of osteoporotic fractures is expected to increase in the coming decades [6, 7]. As osteoporotic fractures tend to occur in a frail older population suffering from comorbidities and polypharmacy, offering geriatric care in the form of CGA to this patient group has repeatedly proven to improve patients outcomes [21]. However, it is well known that there often exists a large time lag between research and routine uptake in daily clinical practice [50]. To increase the sustained uptake of geriatric care in daily clinical practice, this study holds a strong focus on contextual adaptations and refinements, intensive stakeholder involvement and careful selection of implementation strategies tailored to the context and stakeholder needs and preference. By continuously focusing on the local context during the whole implementation phase, the intervention is presumed to be successfully implemented in the immediate phase, holding an increased likelihood of long-term adoption into daily clinical practice [51]. To this end, we blend the design components of effectiveness and implementation studies into a hybrid design, as defined by Curran *et al* [52]. The methodological aspects of the implementation process and its evaluation are out of scope of this paper but will be reported in a complementary paper.

Another important challenge is that the number of experts in geriatric care is not in balance with the geriatric population to deliver care for all older patients [53]. A possible solution to this shortcoming is increasing the geriatric expertise and knowledge of nurses, allied health professionals, and physicians of non-geriatric wards. A transition from a reactive geriatric consultation team model with a one-way approach solely based on input from the geriatric team to a proactive orthogeriatric co-management model with a shared decision making between the multidisciplinary geriatric and surgical team can facilitate diffusion of geriatric expertise to the care teams. In this manner, geriatric care can be embedded into the standard of surgical care to reach more geriatric patients. Moreover, whilst the fracture in itself is the main reason for hospitalization, there is evidence that comorbidities and postoperative complications account for the most impact on morbidity and mortality rather than the fracture itself [54]. As such, a switch from reactive to proactive orthogeriatric care for older people undergoing surgery can improve these postoperative complications [55].

Concerning the high burden of complications occurring in geriatric patients admitted with any type of osteoporotic fracture, there is a need to investigate the effect of orthogeriatric co-management in a heterogenous population [16, 56, 57]. Yet, the body of evidence regarding the effectiveness of orthogeriatric co-management models is mainly based on patients with hip fractures [25, 58, 59]. Patients with non-hip fractures could also benefit from orthogeriatric co-management, however, the evidence is limited [26]. As such, a heterogenous population consisting of community-dwelling and non-community-dwelling patients with all types of major osteoporotic fractures will be included in the G-COMAN intervention and the evaluation in the G-COMAN Fractures study. This heterogenous population represents the geriatric population on the traumatology ward in daily clinical practice.

Since osteoporotic fractures have a high socioeconomic impact related to hospitalizations, it is equally relevant to take the cost-perspective into account [4, 5]. Hence, in-hospital costs and revenues concerning the described intervention implemented in daily clinical practice on the traumatology ward including all hospitalized patients with an osteoporotic fracture will be included in a cost-benefit analysis.

A limitation of this study is the lack of randomization, resulting in a potential risk of selection bias. However, a randomized controlled trial design focusing on targeting care teams to change a model of care might induces a high risk of contamination bias across healthcare professionals. Furthermore, randomized controlled trials do rarely represent real-world data. Therefore, in a single-center context, an observational pre-post study design is considered a better approach for evaluating the effectiveness and implementation of care models [60]. To minimize baseline differences between the usual care and intervention group, propensity score matching will be applied to control for potential baseline confounding. A second limitation of this study is the risk of history bias due to events (e.g., changing of hospital wide procedures also affecting the study population) taking place in between the usual care and intervention group that can affect the study outcomes. To minimize this risk, the time between the usual care and intervention group was limited. A last limitation concerns the hybrid prospective/retrospective aspect of the usual care group. The implementation of a new intervention is a dynamic process that cannot be interrupted while being implemented. Hence, due to the strict time schedule of the implementation process of the G-COMAN intervention, we will not be able to include all 108 usual care patients in a prospective manner. In order to fulfil the sample size to ensure the power of the study, a retrospective usual care group will be recruited on a consecutive base.

In conclusion, The G-COMAN Fractures study wants to prove the beneficial impact of a newly embedded nurse-led orthogeriatric co-management in improving patient outcomes and costs in a heterogenous population in daily clinical practice with high probability of long-term sustainability of the intervention.

## Supporting information

**S1 Checklist. SPIRIT checklist.**
(DOC)

**S1 File. Study protocol.**
(DOCX)

## Acknowledgments

The authors would like to thank the following people for their active participation in the G-COMAN Fractures study and the closely related G-COMAN project: Els Devriendt, Nadja Himschoot, Lore Wellens, Pieter Stivigny, and Tatiana Neeffs.

## Author Contributions

**Conceptualization:** Sigrid Janssens, Marian Dejaeger, Mieke Deschodt.

**Funding acquisition:** Katleen Fagard.

**Investigation:** Marian Dejaeger.

**Methodology:** Sigrid Janssens.

**Supervision:** Marian Dejaeger, Mieke Deschodt.

**Writing – original draft:** Sigrid Janssens, Mieke Deschodt.

**Writing – review & editing:** Sigrid Janssens, Marian Dejaeger, An Sermon, Katleen Fagard, Marie Cerulus, Heidi Cosyns, Johan Flamaing, Mieke Deschodt.

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
