## [Decision Letter · Decision Letter 0]

14 Feb 2023

PONE-D-23-00335Orthogeriatric co-management for older patients with a major osteoporotic fracture: protocol of an observational pre-post studyPLOS ONE

Dear Dr. Janssens,

Thank you for submitting your manuscript to PLOS ONE. After careful consideration, we feel that it has merit but does not fully meet PLOS ONE’s publication criteria as it currently stands. Therefore, we invite you to submit a revised version of the manuscript that addresses the points raised during the review process.

The necessary changes to this protocol all relate to increased clarity:  explicit definitions of important terms and concepts, specification of the statistical hypotheses to be tested, and elimination of ambiguous usage as expressed in either terminology or English language usage.  The authors, having demonstrated considerable thoughtfulness into devising the protocol, should encounter relatively little difficulty in addressing the points raised by the reviewers.

We look forward to receiving your revised manuscript.

Kind regards,

Robert Daniel Blank, MD, PhD

Academic Editor

PLOS ONE

Journal Requirements:

Reviewers' comments:

Reviewer's Responses to Questions

**Comments to the Author**

1. Does the manuscript provide a valid rationale for the proposed study, with clearly identified and justified research questions?

Reviewer #1: Yes

Reviewer #2: Yes

Reviewer #3: Partly

2. Is the protocol technically sound and planned in a manner that will lead to a meaningful outcome and allow testing the stated hypotheses?

Reviewer #1: Yes

Reviewer #2: Yes

Reviewer #3: Partly

3. Is the methodology feasible and described in sufficient detail to allow the work to be replicable?

Reviewer #1: Yes

Reviewer #2: Yes

Reviewer #3: Yes

4. Have the authors described where all data underlying the findings will be made available when the study is complete?

Reviewer #1: Yes

Reviewer #2: Yes

Reviewer #3: Yes

5. Is the manuscript presented in an intelligible fashion and written in standard English?

Reviewer #1: No

Reviewer #2: Yes

Reviewer #3: Yes

6. Review Comments to the Author

You may also provide optional suggestions and comments to authors that they might find helpful in planning their study.

Reviewer #1: This manuscript describes the G-COMAN interventional protocol that explores orthogeriatric co-management involving collaborative geriatric-surgical teams who proactively manage post surgical care of patients >/=75 years of age who presented in the trauma department of University Hospitals, Leuven in Belgium with osteoporotic fractures. Overall, this is a very interesting study and it will be interesting to see the final results of this trial, once published.

Generally, the protocol is clearly detailed and includes expected elements including compliance with the SPIRIT checklist and standard protocol items and use of standardized templates, human subjects approval by institutional review board, trial registration, detailed protocol, reporting of funding source and no conflict of interest, sample size estimates with planned over-enrollment of 10% (although this section might also include some definition of the historical census of study-eligible patients passing through the trauma center on an annual or monthly basis during that time frame, especially given that it overlaps the COVID time frame), detailed documentation of data collection and variables of interest, statistical approaches including the plan for outcome reporting, application of propensity scores to address confounding, Bonferroni-Holm correction related to secondary analysis and approaches to statistical modeling of the data and how missing data and subjects lost to follow-up will be addressed. The introduction lays out the context and purpose of the study well, although the authors may consider couching the study's purpose more clearly as a stated, hypothesis that will be tested in introduction or analytical section (since none is formally stated until the discussion line 388, P22). The discussion also includes study limitations, which include contamination bias. The discussion might also be expanded to include consideration surrounding portability and generalizability of the proposed intervention to other institutions. The authors may also wish to expand a bit on concerns re: ' history bias ' and what outcomes might be impacted, since the statement beginning on line 443 (p.24) is a bit vague.

The main concern relates to question #5 above relating to usage of 'standard English'. Spelling throughout appears to use British English so if American English usage is preferred, this would need to be addressed throughout the manuscript. However, there are some grammatical issues that the authors may wish to address. Since this protocol is already in progress, phases that are already completed should be described consistently in past tense. For example beginning on line 16 (p.6), 'prospective inclusion of patients in the usual care group is limited to five months' (should be 'was limited' since the date range occurs in the past and this is otherwise confusing to the reader. Similarly, on p.7 line 122, a feasibility study will be conducted ...' should read 'was conducted' since it has already been done. The sample size estimation should also be stated in past tense (line 137 p.7), and so on. These are just a few examples, but this needs to corrected throughout the manuscript to help the reader to understand what phases are complete and which remain in progress and in fact, since the 'post' period is almost completed, the entire manuscript should be utilizing past tense including protocol and design of measurement frameworks. Under study participants exclusion criteria, a more precise definition of the what the authors are classifying a 'pathological fracture' would be helpful. For example, I would surmise that osteoporosis was not an exclusion criterion although it would fit the definition a 'pathological fracture'? On line 147 (p. 8) there should be a subheading inserted to labeling that section as 'Feasibility Study' since that phase deserves its own section. Because the discussion of what the feasibility study actually entailed comes much later in the manuscript, the statement re: 'evaluating the feasibility of the intervention' line 150 (p.8) is rather vague, and to help the reader to understand what was actually involved, the authors may consider referring them to Table 2, or ideally, by moving the 'Process indicator variables' text from p.17 to the feasibility section to help the readers' comprehension of the that phase of the study.

To assist the authors, a few typos might also be pointed out. On p. 5 line 86, 'that provides in inpatient geriatric consultation' (remove the word 'in': (i.e. that provides inpatient geriatric consultation'. The wording in the discussion might also be smoothed out a bit and presented in more conventional English. For example, consider removing 'on the long term' (not grammatically correct on p. 23 (line 404) and rewording line 403 to read ...immediate phase, to increase likelihood of long-term adoption into daily clinical practice.' Line 414 (p.23) could be reworded to read: ...multidisciplinary geriatric and surgical team can facilitate diffusion of geriatric expertise to the the care teams. In this manner, ..." To make the sentence on line 446 more grammatically correct, reword as follows: 'We are aware of the limitation introduced by the hybrid prospective/retrospective design of the usual care group.' Again all of the wording in the remainder of that paragraph needs to be converted to past tense, since that phase is complete. On line 455, p. 25, change 'ambition' to 'with high probability of long term sustainability of the intervention'. You may consider splitting that into sentence into 2 two sentences or adding an additional sentence, since you have a 'one sentence paragraph' which is also not grammatically correct, since at least 2 sentences are required to support a paragraph.

Finally, the figures were initially confusing since they are not clearly labeled when the manuscript is printed (only visible online) and it was especially confusing because two of the figures are out of order and what is labeled as 'Fig 1' is actually a table. I suspect that this would be adjusted at the time the manuscript is finalized for publication.

Reviewer #2: The authors present a thorough and comprehensive study protocol for an important study evaluating whether protocolised orthogeriatric co-management of patients with a major osteoporotic fracture may improve in-hospital and post-discharge outcomes. This has the potential to be practice-changing, especially in the local setting.

A few minor points to clarify/add:

- Lines 79-80 – in what diagnoses have the co-management models already been shown to be positive?

- Line 181 – care plans – can you give more details as to what types of care plans will be programmed – physical? Nutritional?

- Line 185 – questionnaire – is this to be filled out by the patient or a caregiver? At what point in the admission will this be done (given that it is about premorbid function and there may be peri-operative/initial delirium)?

- Lines 281-288 – cost-benefit analysis – will this only consider in-hospital costs/costs associated with the in-patient admission, or will it consider some of the post-admission outcomes (eg hospital readmission, mortality)

Reviewer #3: This paper describes the methods being employed in a study of the implementation of an orthogeriatric co-management program for older adults hospitalized with a major osteoporosis-related fracture.

The justification for this study rests on a potentially inaccurate assumption, though that is only a moderate weakness. The authors assert that, "Osteoporotic fractures most often occur in a frail subgroup of geriatric patients having coexisting multimorbidity, polypharmacy, and geriatric syndromes." The two papers cited in support of this assertion do support the association of frailty with polypharmacy and multimorbidity, but not the assertion that most osteoporotic fractures occur within frail older adults. Throughout the manuscript, it appears that the authors are using "frailty," "older adult," and "osteoporotic" synonymously, which I think is an erroneous approach.

Further, though the statistics cited about the consequences of a major osteoporotic fracture are congruent with my understanding of the literature, their writing suggests that the proportions of patients experiencing these outcomes are from among only frail patients. Not all patients who have osteoporosis-related fractures meet a definition for frailty, but they are at risk for having outcomes such as functional decline, decreased QoL, and mortality.

In general, the Introduction makes much about the importance of frailty, but the subsequent methods do not focus exclusively on the frail subset of patients, nor is there any operational definition provided for identifying patients who are genuinely frail.

Evaluation of Feasibility - Some additional detail about how the numerous listed process indicators will be used to determine "if the intervention is successfully implemented as intended" seems necessary. It is otherwise difficult to see how these process indicators map to the goal.

Other than that, my concerns are relatively minor.

1) The target patient population is adults aged 75+ years. A brief justification of this choice would be helpful.

2) Exclusion criteria include: a) polytrauma - identified how?; b) pathological fracture - cancer related? identified how?; c) concomitant joint infection - identified how?

3) Also excluded are patients transferred from another hospital or ward. So if a person hospitalized for another condition falls while in hospital and breaks a hip, they would not be included? Why?

4) Also excluded are patients with an estimated life expectancy <3 months - how is this determined?

5) Risk of in-hospital complications - does this outcome collectively include all complications of interest? Or individual complications? Is there any intent to analyze complications discretely? It is possible that the intervention may have a differential effect on outcomes and be more effective for reducing some outcomes than for others.

6) Outcomes - in-hospital mortality (line 256) - why does this need to come via linkage to the federal database? Would you not otherwise know who has died while in-hospital?

7) Outcome - hospital readmissions - are these "all cause" readmissions or fracture-related. If fracture-related, how is it determined that it is fracture-related?

7. PLOS authors have the option to publish the peer review history of their article (what does this mean?). If published, this will include your full peer review and any attached files.

Reviewer #1: **Yes: **Ingrid Glurich, Ph.D.

Reviewer #2: No

Reviewer #3: No

---

## [Author Response · Author response to Decision Letter 0]

17 Feb 2023

Dear prof. dr. Blank, 

Dear reviewers, 

We would like to thank the editor and the reviewers for the time you have invested in reviewing our manuscript. In order to answer all the questions and comments, we have taken the following actions that can be found in a separate file labeled 'Response to Reviewers': 

• We created a revisions table with three columns: 1) the left column are the reviewers' comments, 2) the middle column are our answers to these comments, and 3) the right column indicates the changes (in bold) made in the manuscript.

• The changes in the manuscript were highlighted using track changes. 

• Two references were added in the introduction section (reference 10 and 11) in order to clarify a statement that was pointed out by one of the reviewers as you will see in the revisions table. 

• A few additional corrections were made by us, these can be found at the end of the revisions table. 

We look forward to hear from you at your earliest convenience. 

Sincerely, 

Sigrid Janssens

---

## [Decision Letter · Decision Letter 1]

13 Mar 2023

Orthogeriatric co-management for older patients with a major osteoporotic fracture: protocol of an observational pre-post study

PONE-D-23-00335R1

Dear Dr. Janssens,

We’re pleased to inform you that your manuscript has been judged scientifically suitable for publication and will be formally accepted for publication once it meets all outstanding technical requirements.

Kind regards,

Robert Daniel Blank, MD, PhD

Academic Editor

PLOS ONE

Additional Editor Comments (optional):

Reviewers' comments:

Reviewer's Responses to Questions

**Comments to the Author**

1. Does the manuscript provide a valid rationale for the proposed study, with clearly identified and justified research questions?

Reviewer #1: Yes

Reviewer #2: Yes

Reviewer #3: Yes

2. Is the protocol technically sound and planned in a manner that will lead to a meaningful outcome and allow testing the stated hypotheses?

Reviewer #1: Yes

Reviewer #2: Yes

Reviewer #3: Yes

3. Is the methodology feasible and described in sufficient detail to allow the work to be replicable?

Reviewer #1: Yes

Reviewer #2: Yes

Reviewer #3: Yes

4. Have the authors described where all data underlying the findings will be made available when the study is complete?

Reviewer #1: Yes

Reviewer #2: Yes

Reviewer #3: Yes

5. Is the manuscript presented in an intelligible fashion and written in standard English?

Reviewer #1: Yes

Reviewer #2: Yes

Reviewer #3: Yes

6. Review Comments to the Author

You may also provide optional suggestions and comments to authors that they might find helpful in planning their study.

Reviewer #1: Overall, your edits to the manuscript have improved the flow of the text. While you have addressed many of the grammatical issues, there are still several sentences where the language is a bit awkward. I have made a few further suggestions to improve readability, which I hope you will find helpful.

On p. 7 line 130: edit text as follows:

'Following core component implementation, intervention group recruitment will be initiated.'

On p. 9 line 190: edit text as follows:

'For example, each of the shift nurses is able to see the care plan regarding'...(continue as written)

On p.20, line 315: would suggest the following:

'Once recruited, outreach to patients via a mailed letter or phone call is initiated to establish communication regarding the post-discharge follow-up procedure(s).'

On p. 21 line 330: the following edit is suggested:

'During the feasibility study, a research assistant recruited patients following hospital admission. Following receipt of written informed consent from the patient or proxy, the assistant performed an in-person interview, documenting subject responses in the baseline case report form and additionally consulted the subject's electronic health record.'

Reviewer #2: Thank you for addressing my comments - no further edits required from my perspective ................

Reviewer #3: Thank you for your detailed and thorough responses to prior comments and suggestions. I am satisfied with the responses and have no further concerns.

7. PLOS authors have the option to publish the peer review history of their article (what does this mean?). If published, this will include your full peer review and any attached files.

Reviewer #1: No

Reviewer #2: No

Reviewer #3: No

---

## [Editor Report · Acceptance letter]

27 Mar 2023

PONE-D-23-00335R1 

Orthogeriatric co-management
for older patients with a major osteoporotic fracture:
protocol of an observational pre-post study 

Dear Dr. Janssens:

I'm pleased to inform you that your manuscript has been deemed suitable for publication in PLOS ONE. Congratulations! Your manuscript is now with our production department. 

Kind regards, 

on behalf of

Professor Robert Daniel Blank 

Academic Editor

PLOS ONE